# Dynamic Changes in Non-Invasive Markers of Liver Fibrosis Are Predictors of Liver Events after SVR in HCV Patients

**DOI:** 10.3390/v15061251

**Published:** 2023-05-26

**Authors:** Paula Fernández-Alvarez, María Fernanda Guerra-Veloz, Angel Vilches-Arenas, Patricia Cordero-Ruíz, Francisco Bellido-Muñoz, Angel Caunedo-Alvarez, Isabel Carmona-Soria

**Affiliations:** 1Department of Gastroenterology and Hepatology, Hospital Universitario Virgen Macarena, 41009 Seville, Spain; paulafer7@gmail.com (P.F.-A.); patricia.corderoruiz@gmail.com (P.C.-R.); flbellido@hotmail.com (F.B.-M.); acaunedoa@gmail.com (A.C.-A.); icarmonasoria@gmail.com (I.C.-S.); 2Department of Gastroenterology and Hepatology, King’s College Hospital, London SE5 9RS, UK; maferguerrita@hotmail.com; 3Department of Preventive Medicine and Public Health, Faculty of Medicine, University of Seville, 41009 Seville, Spain

**Keywords:** hepatitis C virus, new direct-acting antivirals, sustained virological response, liver fibrosis, non-invasive serum fibrosis markers

## Abstract

**Objectives:** The course of progressive liver damage after achieving sustained virological response (SVR) with direct-acting antivirals (DAAs) remains undetermined. We aimed to determine risk factors associated with the development of liver-related events (LREs) after SVR, focusing on the utility of non-invasive markers. **Methods:** An observational, retrospective study that included patients with advanced chronic liver disease (ACLD) caused by hepatitis C virus (HCV), who achieved SVR with DAAs between 2014 and 2017. Patients were followed-up until December 2020. LREs were defined as the development of portal hypertension decompensation and the occurrence of hepatocellular carcinoma (HCC). Serological markers of fibrosis were calculated before treatment and one and two years after SVR. **Results:** The study included 321 patients, with a median follow-up of 48 months. LREs occurred in 13.7% of patients (10% portal hypertension decompensation and 3.7% HCC). Child–Pugh [HR 4.13 (CI 95% 1.74; 9.81)], baseline FIB-4 [HR 1.12 (CI 95% 1.03; 1.21)], FIB-4 one year post-SVR [HR 1.31 (CI 95% 1.15; 1.48)] and FIB-4 two years post-SVR [HR 1.42 (CI 95% 1.23; 1.64)] were associated with portal hypertension decompensation. Older age, genotype 3, diabetes mellitus and FIB-4 before and after SVR were associated with the development of HCC. FIB-4 cut-off values one and two years post-SVR to predict portal hypertension decompensation were 2.03 and 2.21, respectively, and to predict HCC were 2.42 and 2.70, respectively. **Conclusions:** HCV patients with ACLD remain at risk of developing liver complications after having achieved SVR. FIB-4 evaluation before and after SVR may help to predict this risk, selecting patients who will benefit from surveillance.

## 1. Study Highlights

What is known:Achievement of SVR is associated with clinical benefits, including a reduction in hepatic and extrahepatic manifestations.The use of non-invasive scores in patients with HCV infection prior to treatment is well established. Although they may be used to aid disease management after SVR, their application after SVR is uncertain, and is not currently recommended.What is new here:Determining risk factors for LRE after SVR will optimize patient surveillance after therapy.Dynamic assessment of the FIB-4 index, before and after SVR, is an adequate predictor of LRE after SVR in HCV-ACLD patients.This study provides cut-offs for the FIB-4 score one and two years after SVR to predict LRE (portal hypertension decompensation and hepatocellular carcinoma) in HCV-ACLD patients.

## 2. Introduction

Hepatitis C virus (HCV) is one of the most common causes of chronic liver disease worldwide [1]. According to World Health Organization estimates, approximately 100 million people have been exposed to HCV and 71 million had chronic HCV infection [2]. 

In the last decade, the introduction of new direct-acting antivirals (DAAs) has revolutionized the natural history of chronic HCV, achieving sustained virological response (SVR) rates of over 95%, regardless of HCV genotype [3]. However, the residual liver damage that persists in patients with advanced chronic liver disease (ACLD) after SVR has not yet been defined, and the relatively short follow-up after DAA treatments has prevented any firm conclusions from being drawn. 

Post-SVR surveillance is guided by the presence of pre-therapeutic liver fibrosis and the patient’s comorbidities. Liver biopsy is the gold standard for evaluating fibrosis, but it is an invasive procedure with low applicability. Thus, non-invasive techniques, such as transient elastography (TE) and blood-based biomarkers, have replaced this procedure in recent years. While these methods are useful for estimating liver fibrosis before HCV treatment, they are not recommended in assessing liver fibrosis post-SVR [4].

Post-SVR surveillance is currently universally advocated in patients with cirrhosis [5,6], but it is controversial in patients with advanced fibrosis (F3) [6]. 

The aim of our study was to assess factors predicting the risk of liver-related events (LREs) in HCV-ACLD patients who achieved SVR with DAA therapy. Dynamic changes in liver fibrosis after SVR were determined by non-invasive methods to optimize patient follow-up.

## 3. Materials and Methods

### 3.1. Study Population

This was an observational, retrospective, single-center study conducted in Hospital Universitario Virgen Macarena (Seville, Spain). Patients with ACLD who achieved SVR after DAA therapy between 1 November 2014 and 1 December 2017 were included. Follow-up was carried out until December 2020. Advanced fibrosis was defined as TE > 10 kPa. If these measures were not available, patients were selected by serological marker scores: Fibrosis-4 (FIB-4) > 3.25 and/or aspartate-to-platelet ratio index (APRI) > 1.5. Cirrhosis was defined by TE > 12.5 kPa and/or a combination of endoscopy and imaging criteria (presence of esophageal varices and imaging findings of portal hypertension). 

Patient history of hepatic decompensation (ascites, hepatic encephalopathy (HE), spontaneous bacterial peritonitis (SBP) or variceal bleeding (VB)) one year previously, or present at treatment initiation, was collected using electronic medical records. 

Exclusion criteria were (1) age under 18; (2) HCC diagnosis 5 years prior to DAA therapy; (3) Child–Pugh (CP) score > B8; (4) liver transplant; (5) hepatitis B virus (HBV) and/or human immunodeficiency virus (HIV); or (6) patient treated with interferon therapy.

Baseline demographic variables, risk behaviors (past or current drug and alcohol use), medical history (comorbidities such as diabetes, hypertension, dyslipidemia and chronic kidney disease), HCV genotype, DAA therapy, CP class and MELD score were collected. SVR was defined as undetectable HCV RNA at 12 weeks after the end of therapy using the Cobas® 4800 Systems HCV quantitative nucleic acid test (Roche Molecular Diagnostics). During follow-up, abdominal ultrasound and laboratory tests were performed every 6 months. 

### 3.2. Non-Invasive Assessment of Fibrosis and Clinical Outcomes

Non-invasive serological methods to assess liver fibrosis, the APRI ((aspartate aminotransferase [AST] [U/L]/upper limit of normal)/platelet count × 100) [7] and FIB-4 score (age [years] × AST [U/L]/(platelet count [10^9^/L] × alanine aminotransferase [ALT]^1/2^ [U/L])) [7] were calculated within 60 days of prior HCV treatment initiation and during follow-up 1 and 2 years after achieving SVR.

Liver stiffness measurement (LSM) (FibroScan^®^, Echosens) performed within 60 days prior to HCV treatment initiation was recorded. To ensure the quality of measurements, the following criteria had to be fulfilled: (1) at least 10 valid shots; (2) a success ratio (ratio of valid shots to the total number of shots) above 60%; and (3) interquartile range (IQR) ≤ 30%, as recommended by current guidelines [7]. 

LREs were defined as portal hypertension decompensation (ascites, VB, HE) and the occurrence of HCC. Liver-related and non-liver-related mortality was also recorded. Patients were followed until the first occurrence of liver-related decompensation, transplantation or death. Only the first event was reported in a single subject.

### 3.3. Statistical Analysis

For the descriptive analysis, qualitative variables are expressed as absolute frequency (N) and percentage (%). Quantitative variables are expressed as mean (standard deviation, SD) and median (interquartile range, IQR). The normality of quantitative variables was tested using the Shapiro–Wilk test. Comparisons between groups for variables with normal distribution were performed using the Student’s *t* test (for independent samples), and for variables that do not follow a normal distribution using the Mann–Whitney U test. Additionally, the Chi-square test was used to determine the relationship between categorical variables. Continuity correction on 2 × 2 contingency tables was applied, using Fisher’s exact test. 

The Kaplan–Meier curve was used to estimate the cumulative individual survival probability over time. 

Cox regression analyses were performed to identify predictors of liver-related events (LREs). First, univariate analysis was performed to identify statistically significant variables associated with follow-up until the development of portal hypertension decompensation or hepatocellular carcinoma (HCC) (*p* < 0.15). Multivariate analysis was then performed to select the final Cox models to assess the association of each LRE and each variable included in the model controlled by the rest of the variables. The results were expressed as adjusted hazard ratios (HR) and their 95% confidence intervals (CI). 

To investigate the predictive ability of repeated measurement of serological markers to determine the risk of developing LRE, sensitivity and specificity were calculated using a receiver operating characteristics (ROC) curve. 

Analyses were performed using IBM SPSS Statistics 26. 

## 4. Results

### 4.1. Study Population

A total of 573 patients with HCV received treatment with DAAs between 1 November 2014 and 1 December 2017. Of these, 346 had advanced fibrosis or cirrhosis and 336 achieved SVR. Finally, 321 patients met the inclusion criteria and were enrolled in the study (Figure 1). The baseline characteristics of these patients are described in Table 1. Median age was 59.1 (10.6) years and 68.8% (221) were male. Among the main comorbidities, 39.3% (126) had hypertension (HT), 21.5% (69) diabetes mellitus (DM), 13.4% (43) heart disease (HD), 6.5% (21) dyslipidemia (DL) and 4.4% (14) chronic kidney disease (RCD). Past (22.9%) or current (15%) alcohol intake and history of drug use (24.3%) was reported. Most patients had HCV genotype 1 (77.6%).

### 4.2. Follow Up and Liver-Related Events

The median follow-up was 48 (37; 56.0) months. Twenty-two patients died during this period (6.85%), and liver-related mortality occurred in ten patients. Mean follow-up until death was 30 (17.5; 41.0) months.

LREs occurred in 44 patients (13.7%). None of the patients with advanced fibrosis (F3) developed LREs during follow-up. The most frequent was portal hypertension decompensation, which was present in 32 patients (10%). The incidence rate for decompensation was 2.30/100 patient years. HCC occurred in 12 patients (3.7%), with an incidence rate of 0.86/100 patient years. Finally, four patients (1.25%) required liver transplantation. 

### 4.3. Factors Associated with Portal Hypertension Decompensation

The median follow-up until the onset of hepatic decompensation was 48 (35.5; 56.8) months. The baseline characteristics of these patients are shown in Appendix A. 

The type and frequency of hepatic decompensation developed during follow-up are shown in Figure 2.

Univariate analysis revealed that patients who developed portal hypertension decompensations during follow-up had worse liver function according to the CP score, had presented hepatic decompensation before treatment, had higher FIB-4 at baseline and after SVR (1 year and 2 years post-SVR) and higher APRI at baseline and 2 years post-SVR (Appendix A). 

According to multivariate analysis, at baseline, the CP score (hazard ratio (HR) 4.13 [1.74; 9.81]; *p* < 0.001) and FIB-4 (HR 1.12 [1.03; 1.21]; *p* = 0.006) were significantly associated with portal hypertension decompensation. At follow-up, only FIB-4 evaluated at any time (1 year and 2 years post-SVR) was associated with portal hypertension decompensation (Table 2).

### 4.4. Factors Associated with HCC 

The mean follow-up until HCC developed was 28.25 months (SD 4.33; 95% confidence interval (CI) 19.76–36.74). The baseline characteristics of these patients are shown in Appendix A. 

In the univariate analysis, the development of HCC was associated with older age, genotype 3, DM, CP score, FIB-4 score at baseline and after SVR (1 year and 2 years post-SVR) and APRI index assessed 2 years post-SVR (Appendix A). 

According to the multivariate analysis, at baseline, older age (HR 1.13 [1.05; 1.22]; *p* = 0.001), genotype 3 (HR 33.18 [6.24; 176.50]; *p* < 0.001), DM (HR 4.92 [1.26; 19.22]; *p* = 0.022) and FIB-4 (HR 1.15 [1.04; 1.27]; *p* = 0.005) were significantly associated with the development of HCC. At follow-up, FIB-4 remained a good predictor of HCC, 1 year post-SVR (HR 1.38 [1.09; 1.75]; *p* = 0.026) and 2 years post-SVR (HR 1.53 [1.21; 1.95]; *p* < 0.001) (Table 3). 

### 4.5. Implications of Serological Marker Levels after SVR on the Risk of Developing LRE

Dynamic changes in serological marker (FIB-4 and APRI) levels were observed in the assessment before and after SVR, but not on repeated measurements after SVR. This pattern appears both in patients who did not develop any LRE and in those who did (Figure 3).

Additionally, the FIB-4 score assessed at any time during follow-up was associated with the risk of developing any LRE, and therefore we focused on this biomarker. 

Patients who developed portal hypertension decompensation during follow-up also showed a higher FIB-4 score after SVR than patients who did not (1 year post-SVR: 3.9 vs. 1.5; *p* < 0.001/2 years post-SVR: 3.8 vs. 1.5; *p* < 0.001) (Appendix A). Similarly, patients who developed HCC during follow-up had higher FIB-4 levels after SVR than patients who did not (1 year post-SVR: 3.47 vs. 1.56; *p* < 0.007/2 years post-SVR: 4.2 vs. 1.6; *p* < 0.001) (Figure 3). 

In a receiver operating characteristic (ROC) analysis, the FIB-4 cut-offs associated with the risk of developing portal hypertension decompensation were 2.03 (area under the curve (AUC) 0.824; 95% CI 0.754; 0.903) and 2.21 (AUC 0.819; 95% CI 0.733; 0.905) 1 and 2 years post-SVR, respectively. A significant difference in the progression rates of portal hypertension decompensation was demonstrated between patients with FIB-4 ≥ 2.03 1 year post-SVR and FIB-4 ≥ 2.21 2 years post-SVR than those with lower values (*p* < 0.001) (Figure 4).

The FIB-4 cut-offs to predict the risk of developing HCC were 2.42 (AUC 0.732; 95% CI 0.577; 0.886) and 2.70 (AUC 0.811; 95% CI 0.671; 0.951) 1 and 2 years post-SVR, respectively. A significant difference in the rates of developing HCC was observed between patients with FIB-4 ≥ 2.42 1 year post-SVR and FIB-4 ≥ 2.70 2 years post-SVR than those with lower values (*p* < 0.001) (Figure 5). 

Based on these findings, we have proposed an algorithm to guide portal hypertension and HCC surveillance in this population (Figure 6 and Figure 7). 

## 5. Discussion

The long-term development of liver-related complications among subjects with HCV-ACLD after achieving SVR with the new antiviral treatments remains uncertain. In our cohort, 13.7% of the patients developed LREs and 6.85% died during a median follow-up of 48 months. The most frequent LRE was portal hypertension decompensation, which was developed by 10% of the patients, giving an incidence rate of 2.30/100 patient years. HCC occurred in 3.7% of the patients, corresponding to an incidence rate of 0.86/100 patient years. These data follow the same trend as those reported in other international series with larger patient numbers, such as the study by Park et al., with a decompensation incidence rate of 3.2/100 patient years and an HCC incidence rate of 1.7/100 patient years after DAA treatment in the population with cirrhosis [8]. The lower HCC rates in our study could be due to the inclusion of patients with advanced fibrosis (F3). Accordingly, Alonso et al. reported rates of HCC of 0.94/100 patient years [9]. In contrast, Pons et al., during a median 34 months of follow-up, documented an HCC incidence rate of 1.5/100 patient years, but a lower liver decompensation incidence rate, 0.31/100 patient years [10]. In our cohort, portal hypertension decompensation developed later (median time of 48 months). This fact, and the inclusion of patients with previous hepatic decompensation and worse liver function (Child–Pugh B), could explain these differences. 

Our results showed that higher baseline CP scores and a history of previous liver complications were associated with increased risk of the development of portal hypertension decompensation. MELD only reached the limit of statistical significance. These outcomes reflect how the severity of the liver disease before treatment is a determining factor for persistent liver damage after SVR. The large real-world English Expanded Access Program, which also included decompensated patients, confirmed that severity of liver disease before therapy was a predictor of LRE after SVR [11]. A retrospective study by El-Sherif et al., which also included patients with decompensated cirrhosis (CP B or C), found that the presence of previous hepatic decompensation, such as ascites or encephalopathy, was associated with an increased risk of not achieving a reduction in CP to class A after DAA therapy [12]. 

Regarding the non-invasive assessment of fibrosis, patients who developed portal hypertension decompensations also showed more fibrosis, as determined by serological markers. However, only FIB-4 achieved enough statistical power on multivariate analysis to be a good predictor of portal hypertension decompensation at any time during follow-up. Few studies in the literature have evaluated the usefulness of serological biomarkers as predictors of hepatic decompensation after SVR. Although some recent findings support our results, Kuo et al. demonstrated an association between baseline FIB-4 and the risk of developing any type of LRE (portal hypertension complications or HCC) [13], while Takakusagi et al. reported the FIB-4 score before DAA treatment as the only significant factor for the progression of esophageal varices after SVR [14]. 

In relation to HCC, patients who developed this cancer during follow-up were older, had genotype 3 infection and DM comorbidity. The relationship between HCV and DM is well known due to its potential synergism on hepatic damage severity, which accelerates the progression of liver disease. Type 2 diabetes has been found to be a major risk factor for clinical progression in chronic hepatitis, increasing the risk of mortality, cirrhosis, liver decompensation and HCC development among patients without baseline cirrhosis and irrespective of SVR [15,16,17]. Older age has also been related with an increased risk of HCC [18]. The association between genotype 3 and HCC occurrence has also been observed, and may be related to the fact that HCV induces hepatic steatosis, especially in patients infected with genotype 3 [19].

It should be noted that none of our patients with advanced fibrosis (F3) developed HCC or portal hypertension decompensation during follow-up. These results are consistent with data from a recent study with fewer patients (*n* =185) and a shorter follow-up (27.5 months), in which HCC was not reported in F3 patients (TE > 9.5 kPa) [20]. Furthermore, a systematic review and meta-analysis evaluating the incidence of HCC after HCV (including interferon-based or DAA therapy) concluded that the lower incidence found in the F3 group is below the recommended threshold for cost-effective screening [21]. These data are already considered in AASLD guidelines [6] and, if confirmed, will support updates to the current surveillance recommendations. 

With respect to serological markers as predictors of HCC, only FIB-4 achieved the statistical significance to assess its risk at any time during follow-up (before and after SVR). Previous studies have already reported the association between serological markers before DAA treatment and the risk of HCC, especially FIB-4 [17,22,23,24], but also the APRI score [25], and some have even demonstrated the superiority of FIB-4 over the APRI score [26]. 

Although there is no current recommendation to repeat non-serological biomarkers after SVR [4], our results demonstrated the association between FIB-4 assessed at any time after SVR and the risk of LREs. 

Limited data are available on serological markers assessed after SVR as predictors of portal hypertension decompensation. Boursier et al. found that the dynamic course of FIB-4 after SVR was associated with the risk of developing any type of LRE [16], while Yongpisarn et al. also reported FIB-4 after SVR to be a good predictor of the risk of developing LREs [27].

In other studies on HCC reported in the literature, Alonso et al. showed the 1-year FIB-4 score to be an independent factor associated with HCC [9], and Kanwal et al. found a persistent elevation of FIB-4 (>3.25) after SVR to be a good predictor of HCC [28]. Toyoda et al. positively evaluated the predictive capacity of recalculating the FIB-4 score annually after SVR (even after 10 years) for HCC risk and any degree of fibrosis [29]. However, it should be noted that these patients were treated with interferon-based therapy.

Serological markers are easily calculated and provide a lot of information, alerting clinicians to patients who are at higher risk of developing liver complications. For this reason, larger studies should focus on validating these tools after SVR, but also on determining new thresholds post-SVR. 

We found that FIB-4 ≥ 2.03 one year post-SVR and FIB-4 ≥ 2.21 two years post-SVR were associated with a high risk of developing portal hypertension decompensations. After an exhaustive review of the literature, we were unable to identify studies to determine new cut-off values after SVR that can predict portal hypertension decompensations. Instead, thresholds before DAA treatment have been reported. Kuo et al. demonstrated that a baseline FIB-4 ≥ 2.90 was associated with a high risk of portal hypertension decompensation [13], and Takakusagi et al. observed that patients with FIB-4 ≥ 8.41 may have progressions of esophageal varices [14]. 

FIB-4 ≥ 2.42 one year post-SVR and ≥2.70 two years post-SVR were the best predictors of HCC in our cohort. Many studies propose the same thresholds as those used in untreated HCV patients (FIB-4 > 3.25) to predict LREs after SVR [30,31,32]. However, due to the regression of liver fibrosis evidenced after SVR, we consider that lower cut-offs should be defined. There are a few studies that have attempted to validate new cut-off values of non-invasive tools to predict HCC after SVR. Na et al. reported FIB-4 > 2.5 as the best cut-off to predict HCC [33]. Another study, conducted by Kumada et al., associated FIB < 1.50 with a low risk of HCC [34]. These data were similar to those reported by Toyoda et al., who proposed FIB-4 < 1.45 as a good predictor of a low risk of HCC [29]. However, it should be noted that all these cohorts achieved SVR using interferon-based therapies, and thus the results are not entirely comparable. 

Our study has some limitations. It is a retrospective study with all the drawbacks that this entails. A larger number of patients would have allowed us to offer more powerful statistical results in some respects to draw solid conclusions. We had limited data on LSM after SVR, especially in patients who developed LRE during follow-up, so we were unable to make any observations in this regard. Furthermore, due to its retrospective design, we were unable to analyze some risk factors for the clinical progression of fibrosis after SVR, such as body mass index or additional biochemistry data. 

In conclusion, although the incidence rate of LREs decreases after DAA treatment, patients with HCV-ACLD remain at risk of developing these events, even after SVR. Determining the risk factors associated with the development of LREs allows patient surveillance to be optimized. Dynamic assessment of the FIB-4 score, before and after SVR, is a good predictor of LREs after SVR. Further studies with a larger number of patients are necessary to confirm these results, which will allow the selection of patients at higher risk, who would benefit from surveillance. 

## Figures and Tables

**Figure 1 viruses-15-01251-f001:**
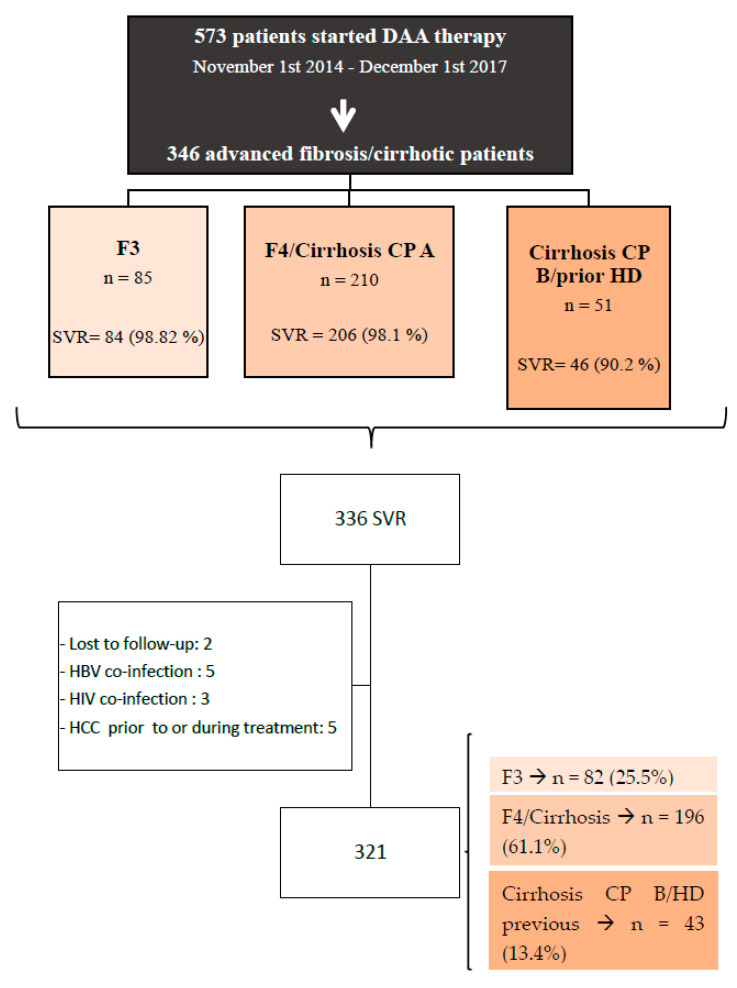
Study flowchart. CP: Child–Pugh; HD: hepatic decompensation; SVR: sustained virological response.

**Figure 2 viruses-15-01251-f002:**
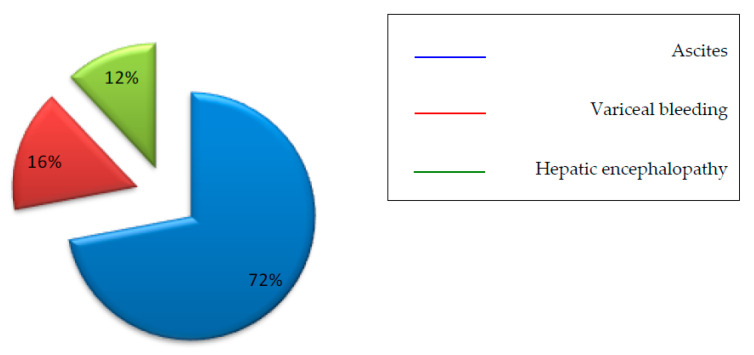
Portal hypertension decompensations developed during follow-up.

**Figure 3 viruses-15-01251-f003:**
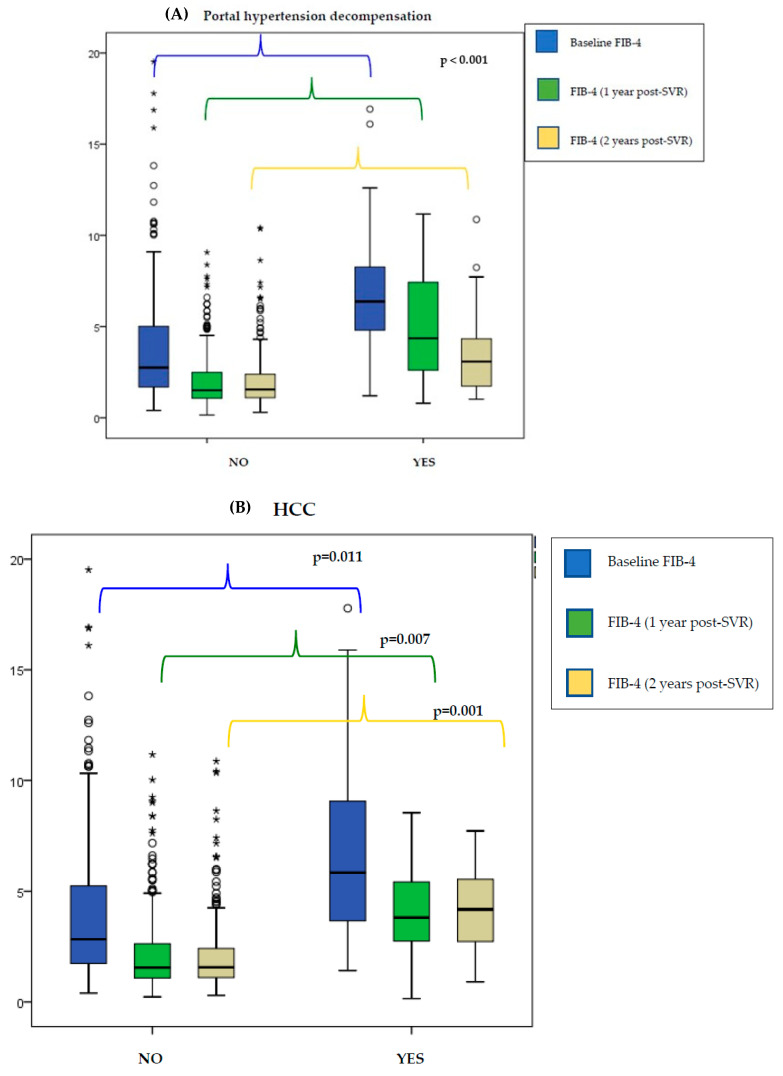
Changes in serological markers during follow-up in patients who developed liver-related events and in those who did not develop them: portal hypertension decompensation (**A**), HCC (**B**). HCC: hepatocellular carcinoma. The circles represent extreme values (>1.5 times the interquartile range). Asterisks (*) represent very extreme values (>3 times the interquartile range).

**Figure 4 viruses-15-01251-f004:**
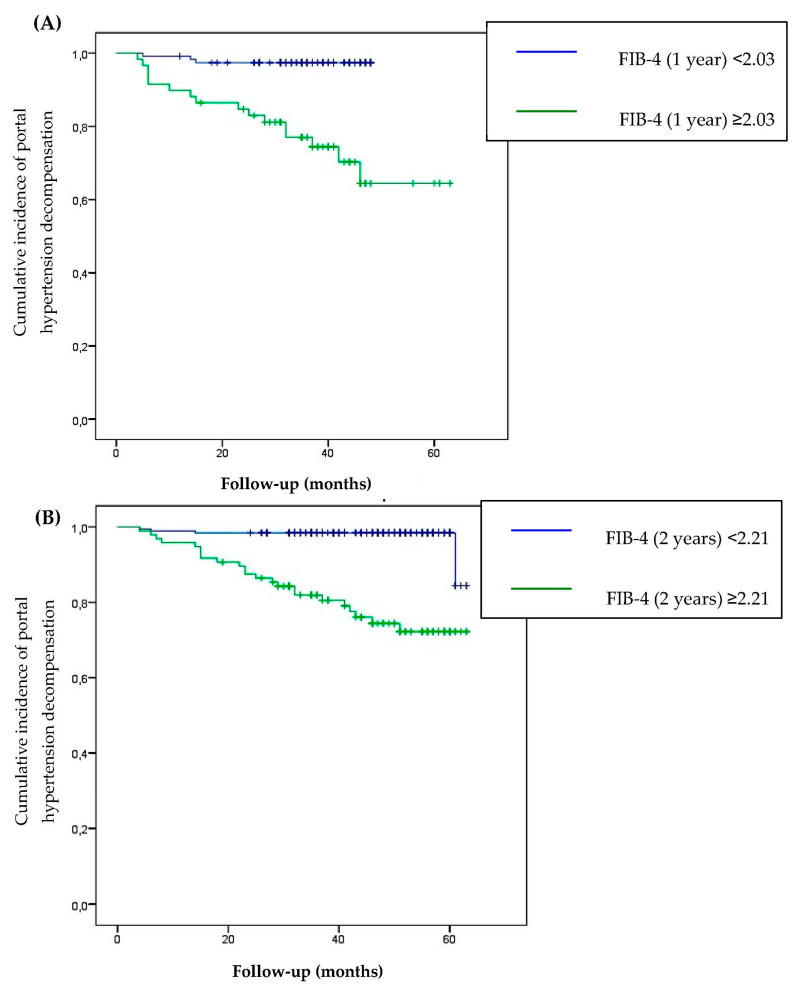
Cumulative incidence of portal hypertension decompensation according to FIB-4 evaluated 1 year post-SVR (**A**) and 2 years post-SVR (**B**).

**Figure 5 viruses-15-01251-f005:**
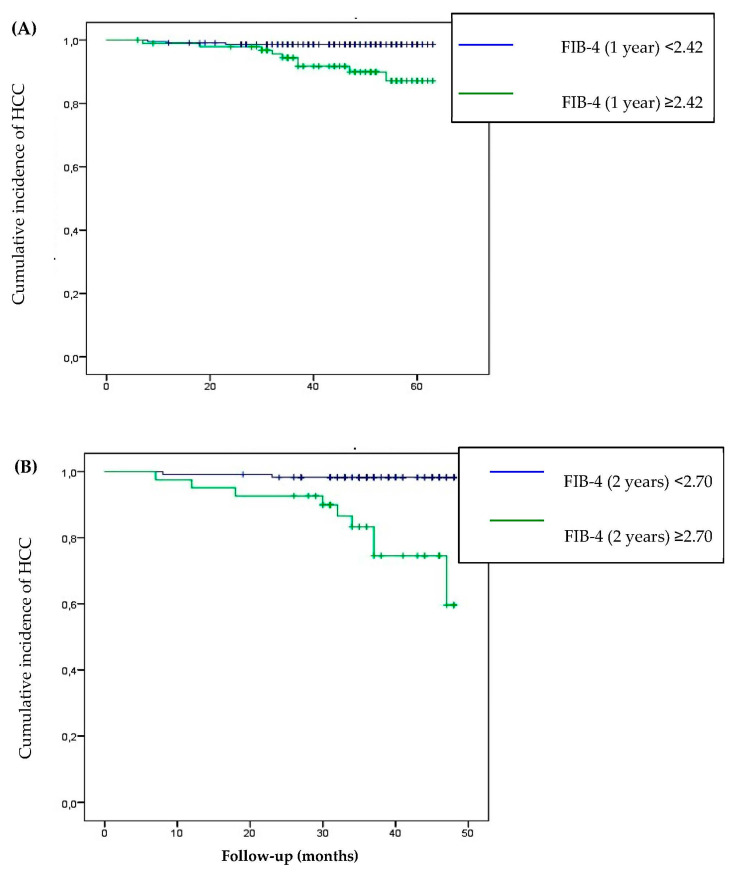
Cumulative incidence of HCC according to FIB-4 evaluated 1 year post-SVR (**A**) and 2 years post-SVR (**B**).

**Figure 6 viruses-15-01251-f006:**
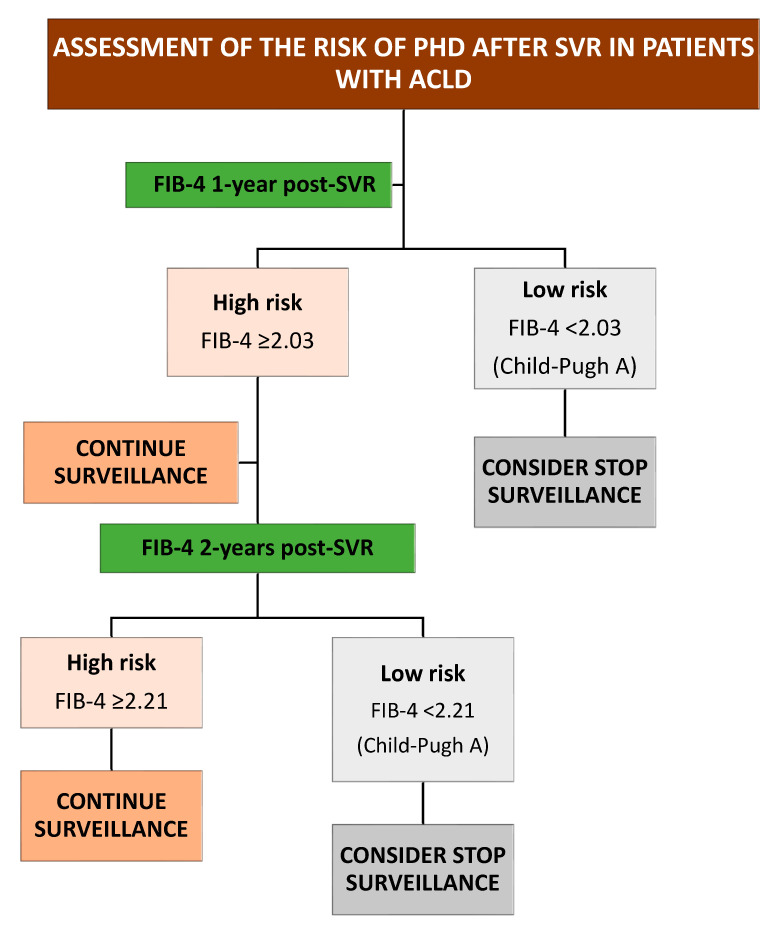
Algorithm to guide portal hypertension decompensation surveillance. ACLD: advanced chronic liver disease; SVR: sustained virological response.

**Figure 7 viruses-15-01251-f007:**
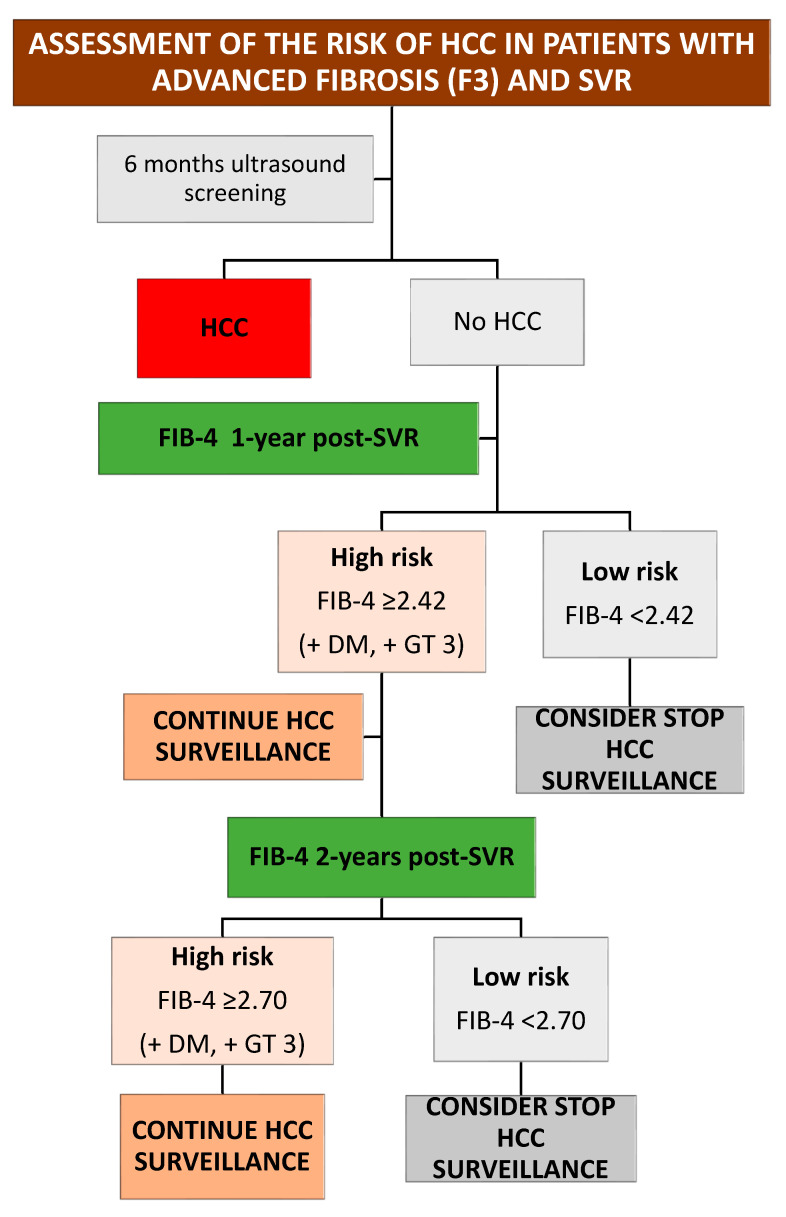
Algorithm to guide HCC surveillance. DM: diabetes mellitus; GT: genotype; HCC: hepatocellular carcinoma; SVR: sustained virological response.

**Table 1 viruses-15-01251-t001:** Baseline characteristics of the patients included in the study.

Characteristics		N (%)/Me	CI (95%)/P_25_; P_75_
Sex	Male Female	221 (68.8) 100 (31.2)	63.8; 73.9 26.1; 36.3
Age		57	52.0; 67.5
Alcohol user	Yes Ex	48 (15.0) 73 (22.9)	11.1; 19.0 18.3; 27.5
IDUs	Yes Ex-IDUs	5 (1.6) 77 (24.3)	0.2; 3.0 19.5; 29.0
HT		126 (39.3)	33.9; 44.6
DM		69 (21.5)	17.0; 26.0
DL		21 (6.5)	3.8; 9.3
CKD		14 (4.4)	2.1; 6.6
HD		43 (13.4)	9.7; 17.1
Genotype	1 2 3 4	249 (77.6) 4 (1.2) 44 (13.7) 24 (7.5)	73.0; 82.2 0.03; 2.5 9.9; 17.5 4.6; 10.4
DAA treatment	1. Sofosbuvir + Daclatasvir 2. Simeprevir + Sofosbuvir 3. Sofosbuvir/Ledipasvir 4. Ombitasvir/Paritaprevir/Ritonavir + Dasabuvir 5. Sofosbuvir/Veltapasvir 6. Elbasvir/Grazoprevir 7. Glecaprevir/Pibrentasvir 8. Sofosbuvir/Velpatasvir/Voxilaprevir	19 (7.6) 30 (12.0) 144 (57.6) 24 (9.6) 9 (3.6) 23 (9.2) 1 (0.4) 0	4.3; 10.9 7.9; 16.1 51.4; 63.8 5.9; 13.3 1.3; 5.9 5.6; 12.8 0; 1.2 0
Child–Pugh	A5 A6 B7 B8	262 (83.7) 15 (4.8) 28 (8.9) 8 (2.6)	79.6; 87.8 2.4; 7.2 5.8; 12.1 0.8; 4.3
MELD		7	7; 8
TE (kPa)		15.4	12.0; 23.0
FIB-4		2.9	1.8; 5.3
APRI		1.2	0.7; 23
Previous decompensation		32 (10.0)	7; 14

APRI: aspartate-to-platelet ratio index; CI: confidence interval; CKD: chronic kidney disease; CP: Child–Pugh; DAA: direct-acting antivirals; DL: dyslipemia; DM: diabetes mellitus; HD: heart disease; HT: hypertension; IDUs: intravenous drug users; Me: median; N: number of patients; TE: transient elastography.

**Table 2 viruses-15-01251-t002:** Factors associated with risk of developing portal hypertension decompensation (multivariate analysis).

		Portal Hypertension Decompensation		
Characteristics		HR	CI (95%)	p
Model 1				
Child–Pugh	A B	1 4.13	1.74; 9.81	<0.001
FIB-4 (baseline)		1.12	1.03; 1.21	0.006
Model 2				
Child–Pugh	A B	1 3.14	1.24; 7.94	0.016
FIB-4 (1 year)		1.31	1.15; 1.48	<0.001
Model 3				
Child–Pugh	A B	1 2.84	1.16; 6.95	0.023
FIB-4 (2 years)		1.42	1.23; 1.64	<0.001

CI: confidence interval; DM: diabetes mellitus; GT: genotype; HR: hazard ratio.

**Table 3 viruses-15-01251-t003:** Factors associated with risk of developing HCC (multivariate analysis).

		HCC		
Characteristics		HR	CI (95%)	*p*
Model 1				
Age		1.13	1.05; 1.22	0.001
Genotype	Others GT 3	1 33.18	6.24; 176.50	<0.001
DM	No Yes	1 4.92	1.26; 19.22	0.022
FIB-4 (baseline)		1.15	1.04; 1.27	0.005
Model 2				
Age		1.14	1.06; 1.23	0.001
Genotype	Others GT 3	1 34.86	6.29; 193.15	<0.001
DM	No Yes	1 4.89	1.21; 19.78	0.026
FIB-4 (1 year)		1.38	1.09; 1.75	0.026
Model 3				
Age		1.10	1.02; 1.18	0.010
Genotype	Others GT 3	1 36.69	6.73; 199.90	<0.001
DM	No Yes	1 6.84	1.59; 29.33	0.010
FIB-4 (2 years)		1.53	1.21; 1.95	<0.001

CI: confidence interval; DM: diabetes mellitus; GT: genotype; HCC: hepatocellular carcinoma; HR: hazard ratio.

## Data Availability

Not applicable.

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
