# Peer review of "Dynamic Changes in Non-Invasive Markers of Liver Fibrosis Are Predictors of Liver Events after SVR in HCV Patients"

_viruses, 2023, doi:10.3390/v15061251_

Round 1

Reviewer 1 Report

This study aimed to study the performance of serum biomarker to predict HCC development among cirrhosis after Hepatitis C treatment with direct-acting antivirals. This study was performed using 321 cirrhotic patients since 2014-2020. The authors nicely performed analysis. They report that FIB-4 values can be monitored as surveillance for HCC development. The study design and results and discussion section are well developed. This paper worth publishing. 

Author Response

Dear Editor,

Thanks for reviewing our manuscript. Please see the response to the reviewer’s comments.

Response to Reviewer 1 Comments

Point 1: This study aimed to study the performance of serum biomarker to predict HCC development among cirrhosis after Hepatitis C treatment with direct-acting antivirals. This study was performed using 321 cirrhotic patients since 2014-2020. The authors nicely performed analysis. They report that FIB-4 values can be monitored as surveillance for HCC development. The study design and results and discussion section are well developed. This paper worth publishing. 

Response 1: Thanks for your comments and for appreciating the manuscript.

Reviewer 2 Report

Important remarks

The authors discussed the potential progress of liver fibrosis after successful antiviral therapy. This problem is currently under intensive analysis to simplify the selection criteria for patients at particular risk of complications or development of HCC. Although the work is another contribution to achieving optimal criteria, and the authors even present algorithms based on their own results, I believe that the proposed proposals are not fully justified.

The study group, on which the analysis was based, is heterogeneous - apart from patients with defined cirrhosis, and even after decompensation incidents, it includes patients with grade 3 fibrosis, which nevertheless affects the results of the median FIB-4 or APRI.

The authors have written that in the group with advanced fibrosis there was no decompensation or development of HCC (? - no information) during the observation period, and therefore this group of patients seems to be „safer”. Maybe it would be worth following up two groups of patients: with and without cirrhosis? Especially that in the Methodology patients are presented separately.

Changes in the coefficients are related to the values ​​before treatment - then the impact of inflammation is important - it seems that to avoid this impact, it would be worth using the results from point SVR12 as a reference point.

Amendments

Tables and figures need to be checked - table captions and headings need to be corrected.

It is unnecessary to repeat the information in the text and table - to be removed from the text.

Interesting figures S1 and S2, worth including in the main text, but after translation into English!

Author Response

Dear Editor,

Thanks for reviewing our manuscript. Please see the response to the reviewer’s comments.

Response to Reviewer 2 Comments

Point 1: The authors discussed the potential progress of liver fibrosis after successful antiviral therapy. This problem is currently under intensive analysis to simplify the selection criteria for patients at particular risk of complications or development of HCC. Although the work is another contribution to achieving optimal criteria, and the authors even present algorithms based on their own results, I believe that the proposed proposals are not fully justified.

The study group, on which the analysis was based, is heterogeneous - apart from patients with defined cirrhosis, and even after decompensation incidents, it includes patients with grade 3 fibrosis, which nevertheless affects the results of the median FIB-4 or APRI.

The authors have written that in the group with advanced fibrosis there was no decompensation or development of HCC (? - no information) during the observation period, and therefore this group of patients seems to be „safer”. Maybe it would be worth following up two groups of patients: with and without cirrhosis? Especially that in the Methodology patients are presented separately.

Response 1: Thanks for your comments and for appreciating the manuscript. We based our results on F3 and F4/cirrhotic patients, analyzing both together, since both groups (F3 and F4/cirrhosis) met the criteria of “compensated advanced chronic liver disease (cACLD)”. It should be noted that patients who had a history of previous hepatic decompensation were compensated at the time of the inclusion, and so were are referring to the new concept defined in Baveno VII as “cirrhosis recompensation”.

We believe that due to the debate of which patients need to continue surveillance after achieving SVR, is it important to analyses the whole group who meet the criteria of cACLD. In this study, we have provided real data based on dynamic changes of non-invasive biomarkers to help the clinicians to decide whether or not patient need to continue with surveillance.

  Point 2: Changes in the coefficients are related to the values ​​before treatment - then the impact of inflammation is important - it seems that to avoid this impact, it would be worth using the results from point SVR12 as a reference point.

Response 2: Thank your comment. Although it is unclear the best liver fibrosis cut-off value to predict liver damage after HCV treatment, all guidelines agree that post-SVR surveillance should be determine by pre-therapeutic liver fibrosis stage. Therefore, values before treatment must be considered the reference point to continue follow up after SVR and that is why we used pretreatment results to classify our cohort of patient.

Due to the impact of inflammation on early fibrosis values after SVR, we did not collect these data. We believe that the rapid improvement in fibrosis values is due to the necroinflamatory activity caused by the viral infection, achieving an early decrease in AST and ALT.

Point 3: Tables and figures need to be checked - table captions and headings need to be corrected.

Response 3: Thanks for your comment. All tables and figures were checking and the mistakes were corrected.

Point 4: It is unnecessary to repeat the information in the text and table - to be removed from the text.

Response 4: Thanks for your comment. All your suggestion were included. We have tried not to repeat the information provided in the tables and text.

Point 5: Interesting figures S1 and S2, worth including in the main text, but after translation into English!

Response 5: Thanks for your comment. We have included these figures in the main text and translate into English.

Reviewer 3 Report

the study is interesting. But the results do not add anything new to what it is already well known. Several studies in the last 3 years confirmed that SVR in cirrhotic patients reduce LRE and LSM, APRI, FIB-4 and other scores or nomograms have been suggested to guide clinicians during the long term follow-up of these patients.

Here some examples:

10.3390/medicina59040814, 10.3390/medicina59030602, 10.1111/jvh.13830, 10.1016/j.dld.2023.01.153, 10.3390/medicina59010146, 10.3390/biomedicines11010166, 10.3390/cancers13153810 etc.

According to these premises, I do not think that this paper is suitable for publication on a journal like Viruses.

Author Response

Dear Editor,

Thanks for reviewing our manuscript. Please see the response to the reviewer’s comments.

Response to Reviewer 3 Comments

Point 1: The study is interesting. But the results do not add anything new to what it is already well known. Several studies in the last 3 years confirmed that SVR in cirrhotic patients reduce LRE and LSM, APRI, FIB-4 and other scores or nomograms have been suggested to guide clinicians during the long term follow-up of these patients.

Response 1: Thanks for your comments. We believe that our study included new real data and highlight the importance of use of non-invasive biomarkers (FIB-4) as a predictor of LRE after SVR. Additionally, we determined cut-off values post-SVR of this non-invasive biomarker with an excellent AUC value (AUC > 0.8) which can easily predict portal hypertension decompensation or HCC development. This important finding could guide the clinicians whether or not patient needs to continue under surveillance. 

As far are we know, we believe that this is the first study that propose new cut-off values for FIB-4 score after SVR with high accuracy.
